# Leaky or polarised immunity: Non-Markovian modelling highlights the impact of immune memory assumptions

**Bastien Reyné** [1,2,3]*, **Tsukushi Kamiya**[4], **Ramsès Djidjou-Demasse**[3,5ʘ], **Samuel Alizon**[4ʘ], **Mircea T. Sofonea**[6,7]*

**1** Univ. Bordeaux, INSERM, INRIA, BPH, U1219, Bordeaux, France, **2** Vaccine Research Institute, Créteil, France, **3** MIVEGEC, Univ. Montpellier, CNRS, IRD, Montpellier, France, **4** CIRB, Collège de France, CNRS, INSERM, Université PSL, Paris, France, **5** École Polytechnique de Thiès, Thiès, Sénégal, **6** PCCEI, Univ Montpellier, INSERM, Montpellier, France, **7** Department of Anesthesiology, Critical Care, Intensive Care, Pain and Emergency Medicine, CHU Nîmes, Nîmes, France

ʘ These authors contributed equally to this work.
* bastien.reyne@inserm.fr (BR); mircea.sofonea@umontpellier.fr (MTS)

**Data availability statement:** Code is available on Zenodo or GitLab https://gitlab.com/ reyb/immunity-leaky-vs-polarised; https://doi.org/10.5281/zenodo.14620451.

## Abstract

Mathematical models tend to oversimplify the biological details of vaccine or infection-derived immunity effectiveness. Yet, epidemiological outcomes may diverge when assuming polarised immunity—individuals are either fully susceptible or completely immune—compared to leaky immunity—where all individuals are partially protected. We explore the differences between the two by taking advantage of a non-Markovian framework, which allows us to explicitly record the 'age' of the immunity and vary its effectiveness accordingly. A basic scenario reveals that leaky immunity leads to a shorter time between reinfections. A more data-driven scenario based on SARS-CoV-2 data finds that leaky immunity yields substantially more reinfections than polarised immunity and a higher number of infected individuals, yet with a lower probability of hospitalisation. Our findings emphasize the critical role of immune memory modelling assumptions, especially for long-term epidemiological dynamics and public health policies.

## Author summary

Epidemiological models that incorporate immunity often rely on strong assumptions about immune memory, yet the assumptions are rarely acknowledged or justified. Using a non-Markovian framework, we explore the implications of choosing leaky immunity (all hosts are partially protected) over more commonly assumed polarised one (some hosts are fully protected and the others fully susceptible), We find that leaky immunity comes with a shorter time between reinfections and alters estimations of some metrics such as the number of reinfections.

**Funding:** RDD and MTS acknowledge funding from the ExposUM Institute of the University of Montpellier (funded by the Agence Nationale de la Recherche, ANR-21-EXES-0005, part of the France 2030 programme, and by the Occitanie Region; NEXUS EMIPSA project). MTS also acknowledges funding from the ANRS Maladies Infectieuses Émergentes and France 2030 (ANRS-24-PEPRMIE-0003 PReViX project). The funders had no role in study design, data collection and analysis, decision to publish, or preparation of the manuscript.

**Competing interests:** The authors have declared that no competing interests exist.

# 1. Introduction

The early stages of the management of the COVID-19 pandemic faced uncertainties about the natural history of SARS-CoV-2 infection and subsequent immunity. Quite rapidly, estimates were published for the infection fatality ratio (IFR) [1], the generation time [2], and the basic reproduction number [3]. These key metrics helped to parameterise mathematical models, which were instrumental in informing short-term health policies. In particular, mathematical models were used to help determine the intensity of non-pharmaceutical interventions (NPIs) needed to contain the epidemic to avoid an overload of the healthcare system [4,5]. Less than a year later, while the first vaccines were being deployed, the arrival of the first variant of concern (VOC), Alpha, brought new uncertainties to epidemiological modelling given its increased transmissibility and virulence [6–9]. Later, long-term surveillance efforts demonstrated that the effectiveness of vaccination or infection-derived immunity tends to decline over time [10,11]. Notably, in late 2021, the emergence of Omicron VOCs with increased transmissibility, due to their strong immune escape properties, rendered the immunity component paramount in mathematical models [8]. By this time, however, models were already parametrised and ran in routine, and this new evidence regarding immunity was implemented on top of already existing models, leading many modellers to keep the initial assumptions regarding immunity. These initial assumptions often assumed perfect natural immunity, or imperfect immunity with a constant (over the age of immunity) increase in susceptibility relative to Omicron or an increased loss rate of immunity. This (lack of) justification for the assumptions regarding immunity in SARS-CoV-2 models was mostly due to the urgency of the context and the need for timely short-term projections. Indeed, these assumptions did not require changes in the core structure of the model as the differences would have been negligible over such short periods. However, these additional assumptions obscured competing views on the functioning of immunity, which few studies acknowledged (but see [12]).

There is a dense literature on modelling imperfect immunity, as this problem arises for many pathogens. Some of the earliest examples are variations of the well-known *Susceptible – Infected—Recovered* (*SIR*) model that relaxed the assumption of perfect immunity. For example, it is the case of the *SIS* model [13]—where individuals become susceptible again immediately after an infection—and the *SIRS* model [14], where perfect immunity waned over time. These early studies focused on mathematical properties and asymptotic behaviours of the dynamical systems and laid the foundation for later work specific to particular pathogens, such as gonorrhoea for the *SIS* model [15] or syphilis [16] and influenza [17] for the *SIRS* model.

Some statistical approaches were also developed to quantify the effectiveness of imperfect immunity [18]. Such an effort was particularly prolific in the context of vaccination where vaccine trials were specifically designed to estimate the vaccine effectiveness. In their review, [19] elaborated the different interpretations and conceptions behind what appears at first as a simple estimation of vaccine efficacy. Indeed, two competing viewpoints are brought to light. In the first one, the vaccine is assumed to be *leaky*, meaning that it protects everyone homogeneously but imperfectly to a certain degree. In the second, the vaccine is assumed to protect only a fraction of the individuals but with perfect protection. We refer to this all-or-nothing pattern as *polarised*, following a terminology introduced by [20]. For a level of immunity $x \in [0, 1]$ at a given time in the population that presents the same effectiveness, the leaky formulation assumes that each individual benefits from a decrease in susceptibility of $x$, whereas the polarised one assumes that $x$ individuals are perfectly immune and $1-x$ are entirely susceptible (Fig 1A). Both of these formalisms are consistent with the definition of

vaccine effectiveness [21], which is defined as the ratio of incidence rates between vaccinated (immunised) and unvaccinated (susceptible) individuals. Both are also valid implementations of the (ill-defined) term of *imperfect immunity* in mathematical language. Therefore, this lack of precision begs the question: 'Does imperfect immunity equate 1) full protection only for some or 2) partial protection for all?' Note these two presented formalisms could be seen as particular cases of a more general pattern that could account both for individuals perfectly or partially immune and thus may overlook other particular cases such as immune boosting where exposure may reinforce immunity [20].

One difficulty in choosing between these two points of view comes from the lack of a direct comparison between them. To determine whether immunity is leaky or polarised at a given time after recovery, we must track the duration spent in the recovered compartment and specify the corresponding immunity level. This requirement calls for a non-Markovian assumption of immune memory, in which the infectious and immunological history of an individual (*e.g.* how long they have been recovered for) affects the probability of occurrence of some events (*e.g.* the probability to be reinfected the next day).

In this study, we aim to highlight the main differences and the intricacies implied by assuming a leaky or a polarised immunity to model imperfect immunity (Fig 1B). Some of the vast data gathered during the SARS-CoV-2 support the hypothesis that the immunity to this specific pathogen is leaky. In particular, [22] reached this conclusion by using a closed-setting (correctional facilities) to assess the reinfection risk while estimating the overall temporal exposure. However, nothing guarantees this to be a general rule for every pathogen as controlled reinfection data remain rare. Building on an existing mathematical modelling framework to explicitly track the time since the recovery (*i.e.* the age of immunity) [5,23], we perform rigorous comparisons between the two viewpoints, with the flexibility to assume any specified function representing the decline of immunity over time. For the polarised assumption, the *SIRS* model is the simplest formulation under this assumption. For the leaky assumption, we postulate that a recovered individual can be reinfected directly without going back into the susceptible compartment $S$ [24]. The path followed by an individual is then $S \rightarrow I \rightarrow R \rightarrow I$, so we refer to this model as *SIRI* (Fig 1C). We first highlight the differences between the two formalisms using simple models, before assessing the practical issues arising in a specific, data-driven setting by fitting the first SARS-CoV-2 Omicron epidemic waves, for which there are reliable estimations of the level of imperfect immunisation.

## 2. Results

We first compare the time spent uninfected in the leaky and polarised formalisms in a simple discrete-time setting where we assume a constant force of infection and the same decreasing function $\xi(\cdot)$ that represents the immunity efficacy and depends on the age of immunity. We assume all individuals gained immunity at the same time ($\tau = 0$), and we study the proportion that is reinfected over time. Under the leaky immunity assumption, $\xi(\tau)$ represents the decrease in susceptibility $\tau$ days after acquiring immunity. The proportion of remaining uninfected individuals ($q_\ell$) at time $\tau + 1$ is

$$q_\ell(\tau + 1) = q_\ell(\tau) - \underbrace{q_\ell(\tau)[1 - \xi(\tau)]\Lambda}_{\text{newly infected individuals}} = q_\ell(\tau) - \Lambda q_\ell(\tau) + \Lambda q_\ell(\tau)\xi(\tau), \qquad (1)$$

where $\Lambda$ represents the force of infection.

Under the polarised immunity assumption, $\xi(\tau)$ represents the proportion completely immune at time $\tau$. We may decompose the proportion of remaining uninfected individuals

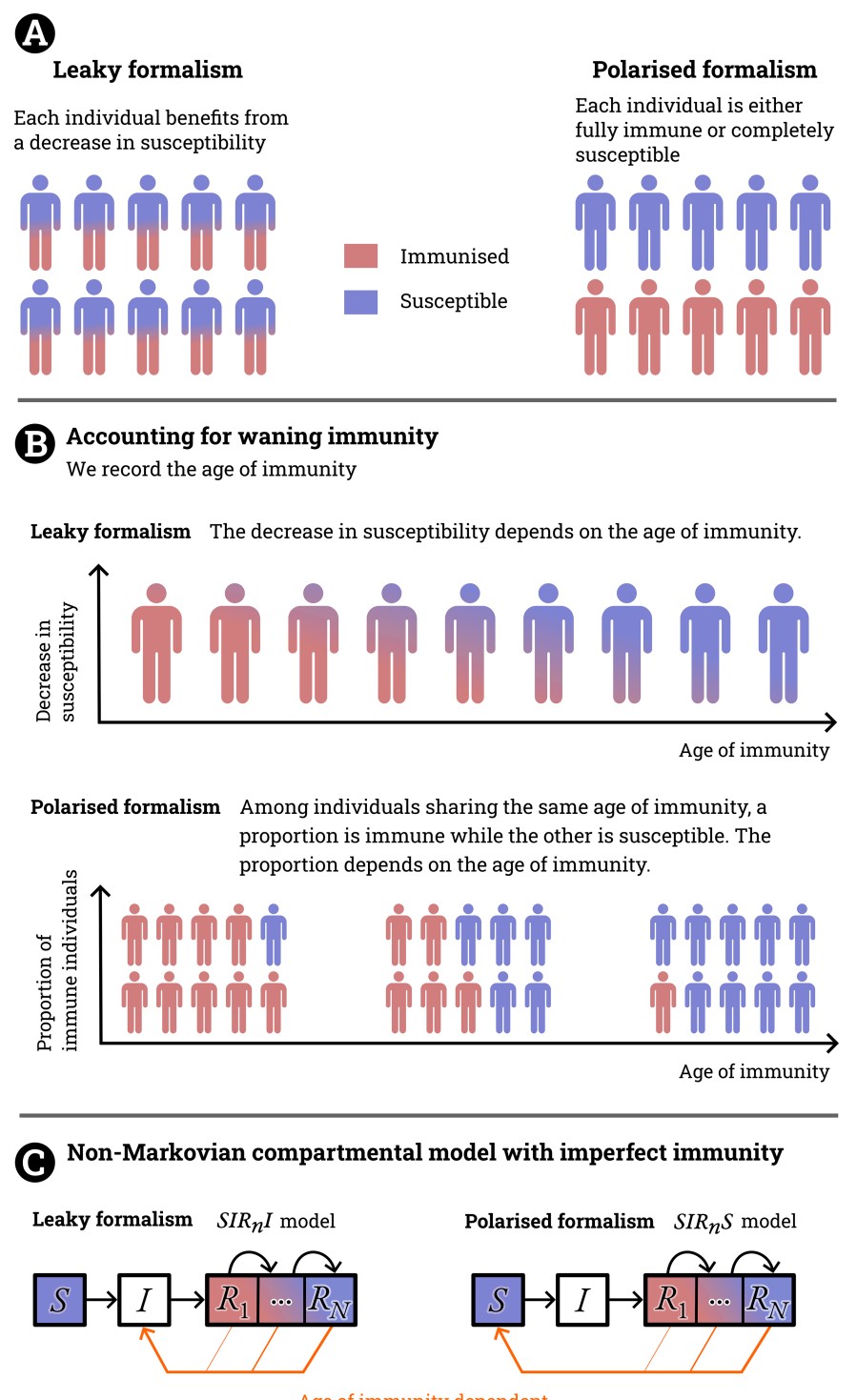

**Fig 1. Illustration of the leaky and polarised formalisms and how they can be modelled. A.** Differences between leaky and polarised formalisms. **B.** How this distinction interacts with immunity waning. **C.** Simplest non-Markovian formulation of the *Susceptible – Infected – Recovered* to incorporate imperfect immunisation for each formalism.

$(q_p)$ at time $\tau$ as $q_p(\tau) = \xi(\tau) + s(\tau)$, where $\xi(\tau)$ (resp. $s(\tau)$) represent the completely immune (resp. susceptible) individuals at time $\tau$. The proportion of uninfected individuals at time $\tau + 1$ is given by

$$
\begin{aligned}
q_p(\tau + 1) &= \xi(\tau + 1) + s(\tau + 1), \\
&= \xi(\tau + 1) + \underbrace{s(\tau)[1 - \Lambda]}_{\substack{\text{susceptible not infected} \\ \text{among already susceptible}}} + \underbrace{[\xi(\tau) - \xi(\tau + 1)]}_{\text{new susceptible}}, \\
&= [s(\tau) + \xi(\tau)] - \Lambda s(\tau), \\
&= [s(\tau) + \xi(\tau)] - \Lambda[q_p(\tau) - \xi(\tau)], \\
&= q_p(\tau) - \Lambda q_p(\tau) + \Lambda \xi(\tau).
\end{aligned}
\tag{2}
$$

We assume $q_\ell(\tau = 0) = q_p(\tau = 0) = 1$. As detailed in Sect A.1 in S1 Text, it can be shown that $q_\ell(1) = q_p(1)$ and $q_\ell(2) \leq q_p(2)$. By induction, we may extend this inequality beyond time $t = 2$, to any time by assuming it to be true at time $t = \tau$, (meaning $q_\ell(\tau) \leq q_p(\tau)$), this inequality remains true at time $t = \tau + 1$ thanks to Eqs (1)–(2). This proves individuals are reinfected more quickly through leaky than polarised immunity—assuming the function $\xi(\cdot)$ is decreasing. This can be further demonstrated numerically with a variety of decreasing functions $\xi(\cdot)$, as shown in Fig 2.

This comparison highlights the divergent outcomes when assuming leaky versus polarised immunity, however, the application remains restricted to cases where the force of infection is known. To explore the implications over time with multiple reinfections in a dynamical setting, when the differences between formalisms accumulate over time, we implement a more realistic scenario capturing the dynamics of the first Omicron epidemic waves on hospital admissions in France. In particular, we focus on the impact of the choice of immunity on the inference of key epidemic parameters, such as the probability of being hospitalised following an infection. The transmission models for each paradigm are illustrated in Fig 3. The main difference is that recovered compartments are followed by a susceptible one in the polarised formalism whereas recovered individuals in the leaky paradigm are reinfected directly. Additional details can be found in the Methods and models section.

The model assuming either form of immune memory could effectively capture the hospital admissions dynamic (Fig 4A). However, differences are observed in key epidemiological outcomes. Since the leaky formalism induces shorter times before reinfections than the polarised one, it exhibits a greater cumulated number of reinfections over time (Fig 4B). From the emergence of Omicron in France (November 26th, 2021) to the end of the monitoring (January 31st, 2023), this corresponds to a difference of 4 million reinfections (and a wider uncertainty). The prevalence is also higher in the leaky model than in the polarised one at all times (Fig 4C). Finally, there are also differences in terms of the force of infection (Fig A in S1 Text) and hospitalisation probability (Fig 4D).

These results hold for different transmission rates, timespans as well as different initialisations for the $R^D$ compartment, as shown by sensitivity analyses (Sect G in S1 Text).

## 3. Discussion

Immunity is rarely perfect in that some people contract infections when others do not under the same circumstances of exposure to infection. However, this phenomenon can correspond to different immunological realities: *i.e.*, all individuals can be partially protected (the 'leaky' paradigm), or some of the individuals may be fully immune while others are fully susceptible

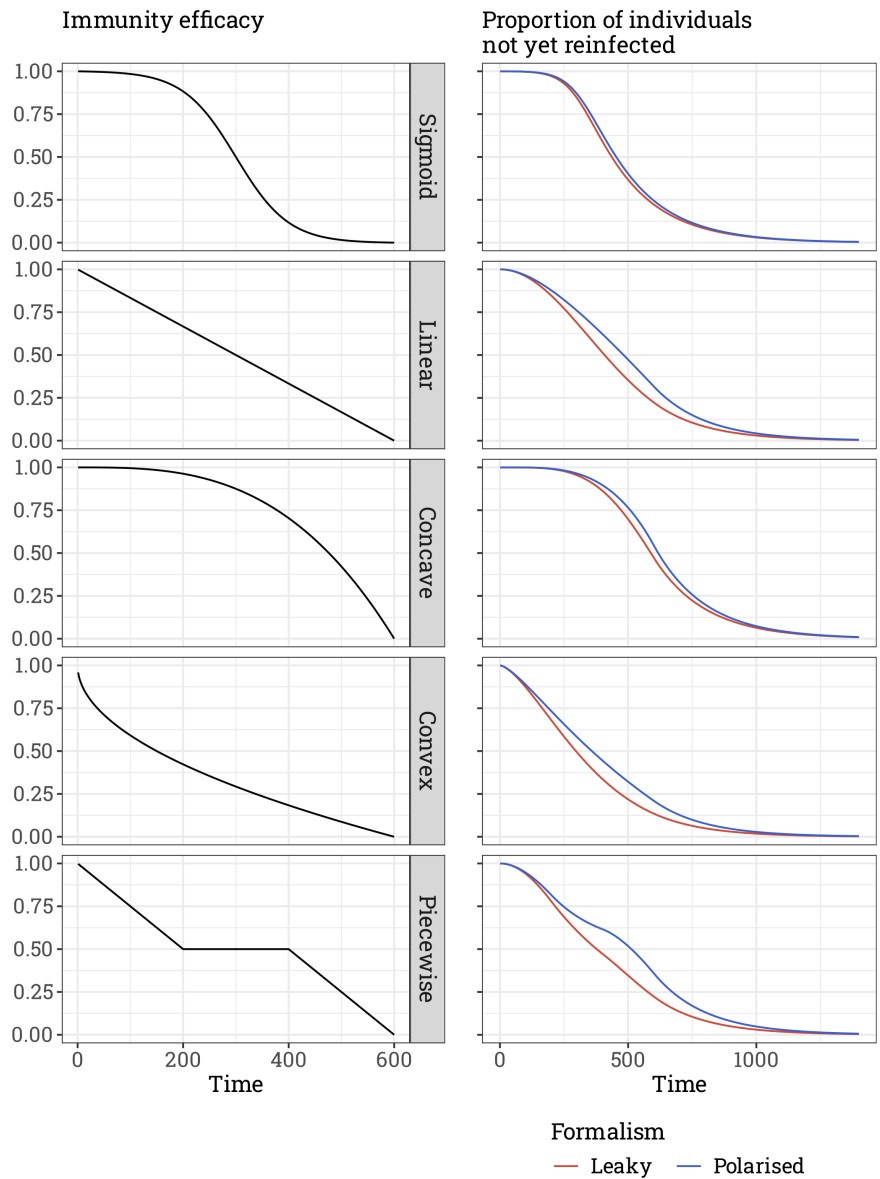

**Fig 2. Time spent uninfected with the leaky and polarised formalisms (on the right) given the imperfect immunity function (on the left).** The force of infection is assumed to be constant ($\Lambda = 0.005$) and all the other parameters are default. More details in Sect A in S1 Text.

(the 'polarised' paradigm) among the most popular formalisms. This study aimed to highlight the methodological and dynamical differences of modelling imperfect immunity, and demonstrate implications for epidemiological outcomes. From a mathematical standpoint, we showed that a leaky immunity generates a shorter average time between reinfections in the population than a polarised one.

Using a data-driven scenario of the epidemic of the Omicron variant of the SARS-CoV-2 on a medium-term scale, we can corroborate a greater number of reinfections over time when assuming leaky immunity (Fig 4B). Since we focused on a limited period, the absolute number of SARS-CoV-2 reinfections is likely to be underestimated. We may also notice an overall

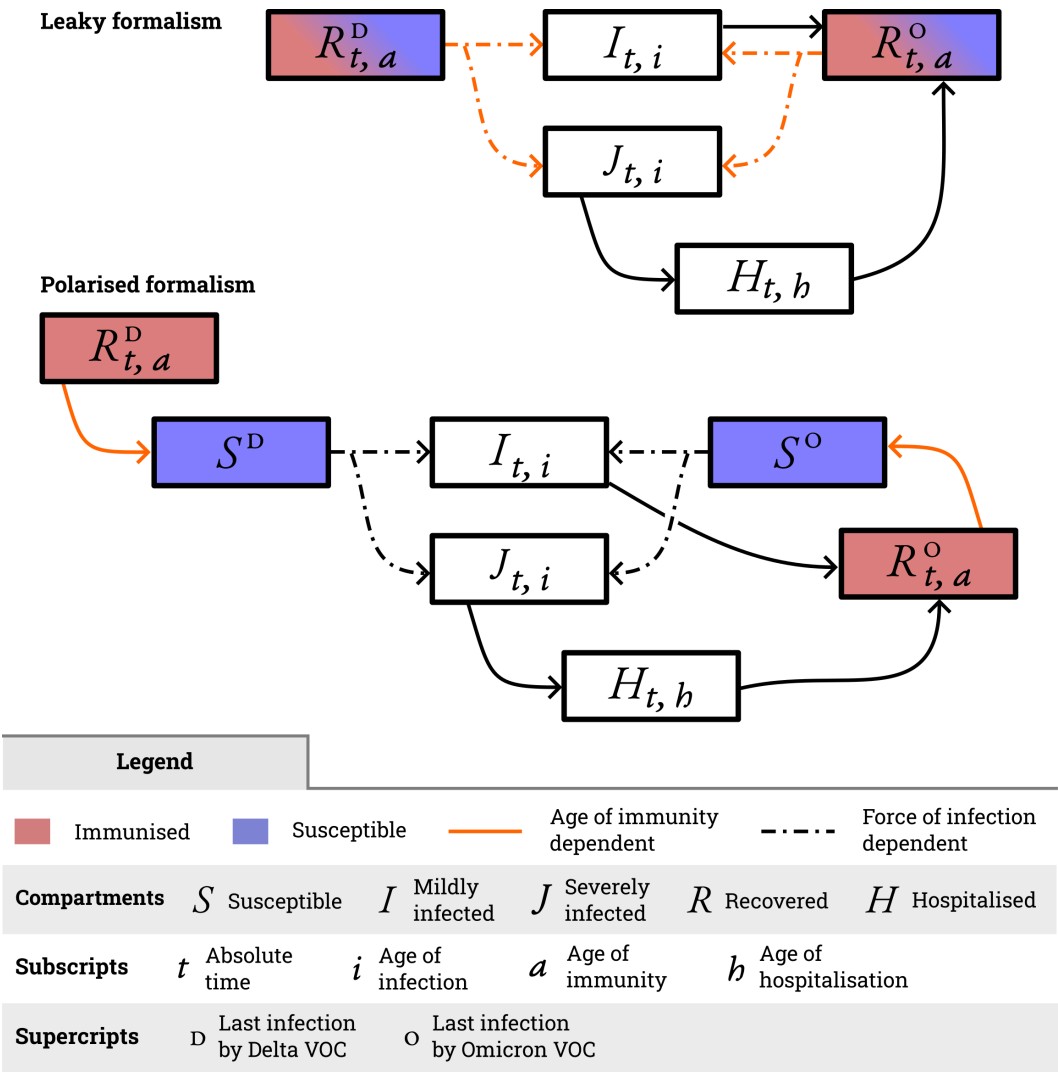

**Fig 3. Structure of the Omicron epidemic waves models.** The model assumes imperfect immunity that acts as an age-dependent decrease in susceptibility in the leaky model and an age-dependent departure rate from the recovered compartment in the polarised one. Before the emergence of the first Omicron VOC, we assume everyone has been immunised for SARS-CoV-2 (past infection or vaccination).

higher number of infected individuals with leaky immunity perhaps because shorter immunity induces more individuals to be reinfected sooner. This in turn could further increase the force of infection and thus sustain a higher level of infected individuals. However, since this is balanced out by a lower probability of hospitalisation, we observe a similar number of hospitalisations in both formalisms. Because the force of infection (through the transmission rates) is fitted to reproduce observed hospitalisation data in both cases, it is tempting to conclude that the differences are negligible from a practical point of view and have little impact on the produced results in some cases. However, such an assertion neglects the key role of the number of re/infections, and its documented impact on long-COVID, next epidemic waves projections, or the probability of the emergence of new strains. Note also, that we made some

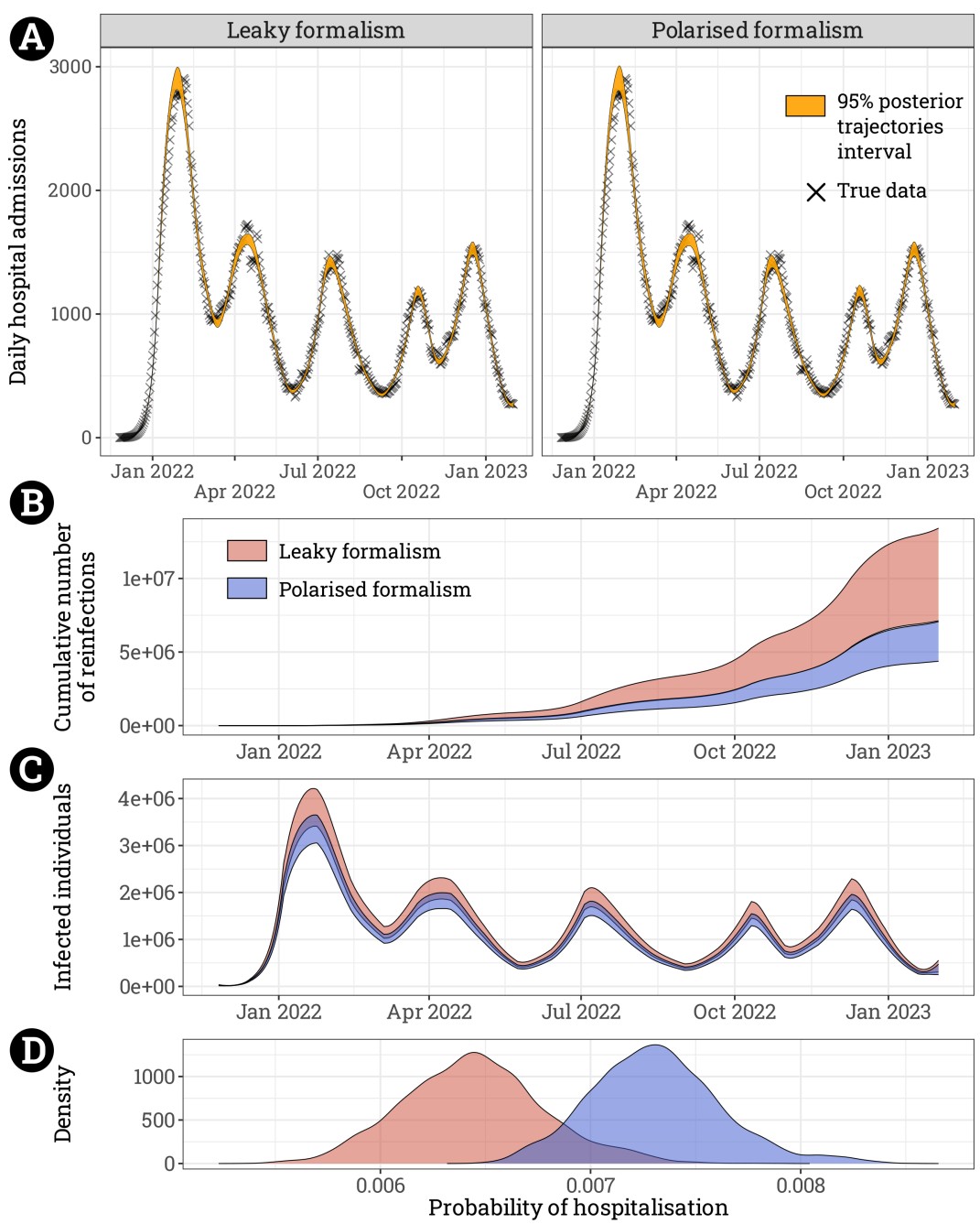

**Fig 4. Comparison of the leaky and polarised formalisms applied to Omicron epidemic waves. A.** Fit of the Omicron epidemic waves with both models. Black crosses represent daily hospital admissions in France. **B.** 95% posterior trajectories interval of the estimated cumulative number of reinfections and of the **C.** number of infected individuals per day. **D.** Posterior probability of hospitalisation following an infection.

simplifying assumptions that might have been questioned in a real-life applied case. In particular, we assumed homogeneity in the population both in terms of susceptibility and contact where there is a difference, in reality, [1,23]. Although overall trends related to immunity and attack rates were similar across age groups in countries like France [25], further studies

explicitly accounting for the age structure of the population are needed to effectively assess its impact in the setting developed here. We also neglect death in our model, which is not linked to the choice of the immunity formalism, although it could have marginally diminished the number of reinfections. We neglected the Omicron sub-lineages that could carry different phenotypic traits (and possibly some immune evasion properties) that have been absorbed in the transmission rates. Finally, we pooled all immunological profiles on a unique recovered compartments that account for all types of immunity (infection derived immunity, different vaccination profiles), and only took into account the age of immunity as factor influencing the efficacy. Beyond that, other assumptions such as the initialisation of the initial level of immunity or the choice of the timespan for the transmission rates have little impact on the results (*cf.* Sect G in S1 Text).

Nevertheless, the differences we identify remain negligible in the short term (at least in the case of SARS-CoV-2, where hospital admissions could be well predicted up to five weeks [26]) since immunity intervenes on longer time scales (here, a year without NPIs). This implies the difference arising in the choice of the immunity paradigm can go unnoticed in the short term—even if we may observe some differences in the projections made, see Sect H in S1 Text. Worst, it could remain undetected in cases where the transmission rate is fitted periodically to provide short-term projections (as in [26] for the SARS-CoV-2 pandemic). The long-term effects on reinfections are difficult to anticipate, but a plausible consequence would be an underestimation of the number of recovered individuals if the "true immunity" lies closer to the leaky formalism than the polarised one, as [22] suggests. Both assumptions were made in the SARS-CoV-2 literature. For example, [12] used the leaky paradigm, [27] used the polarised formalism, and [28] tried both approaches. We may also wonder about the underlying motivations for some studies that assumed a polarised immunity for circulating strains, but a leaky one for new strains evading previously built immunity [29,30]. Eventually, [22] provided real-life observational evidence in favour of leaky immunity.

Our results follow a long list of previous modelling studies that tried to capture imperfect immunity using compartmental models [31–34]. However, the insights from these earlier works remained theoretical because they used only one compartment to model (partly) immune individuals (vaccinated or recovered). Our results also complement other historical works that investigated how to estimate vaccine effectiveness from clinical studies early on [18,19,21] that led the way to assess the level of immunity waning. Most of the estimations of immune waning are made using antibody titres, as in the case of SARS-CoV-2 [35,36], or with a focus on the link between antibody titres and the time between reinfections in the case of other coronaviruses such as 229E and OC43 [37,38]. However, antibodies are one component among many involved in the functioning of immunity, and the link between antibodies and the level of protection is not straightforward. Still, assuming that they can serve as correlates of protection, they can highlight the impact of the individual-level variability in the immune response, with sometimes a high level of dispersion in the immunity efficacy estimations [39]. In our study, we made a simplifying assumption that disregards any forms of heterogeneity other than the time-acquisition declining immunity. This helps have a clearest overall behaviour and pattern, but it is not well-suited to modelling reinfections of weakly protected individuals.

On the methodological level, non-Markovian models are convenient to account for medium-term immunity waning as they allow us to explicitly parameterise the level of protection, whether assuming leaky or polarised immunity. In this study, we explored both assumptions using a discrete-time approach (following [5]): advantages of this approach and alternatives(*e.g.,* partial differential equations [23,40]) are detailed in Sect B.3 in S1 Text.

There are also other attempts in the literature. For instance, in the line of the theoretical study by [41], some studies use the 'chain trick' on recovered compartments to add some heterogeneity in the imperfect immunity without departing from the classical ODE framework [42–44]. There are also less conventional ODEs-based models, such as the ones based on hyper-exponential waiting distributions [45], where individuals may enter different compartments (with some probability), each compartment having its waiting time distribution. This results in an overall distribution with a standard deviation higher than its expectation (hence the name hyper-exponential). This may present an interesting alternative to hypo-exponential distributions when dealing with highly variable biological distributions.

Part of the challenge in modelling imperfect immunity stems from the numerous unknowns surrounding the underlying mechanisms. One possibility to cope with this uncertainty is to adopt a flexible framework that allows for continuity between leaky and polarised immunity [20,46] (and see Sect I S1 Text, for a simple hybrid model). Other possibilities may include allowing a fraction of individuals to recover and the other fraction to go back straight into the susceptible compartments (*e.g.* immunocompromised individuals). [20] also consider the possibility of immune-boosting mechanisms where exposure could reinforce individual immunity even if infection fails to establish. The level of data granularity we would need to provide population estimates is beyond immediate reach, as it would imply a controlled setting and be able to control exposure on individuals which is ethically questionable. Nonetheless, accounting for this hypothesis may already provide some new information, as in the case of [47] where they provide counterfactuals on the pertussis prevalence according to various levels of immunity boosting. Complementary to the immune boosting theory, [48] modelled the immune response given different inoculum sizes and observed non-linear and non-monotonic infection duration, which may open perspectives to linking the level of exposure with a given outcome — infection or boosted immunity.

In the meantime, statistical estimations of the type of immunity through closed settings [22] and temporal meta-analytic estimations of the immunity effectiveness [11], may provide a good starting point in modelling epidemics at a medium-term scale. We acknowledge that our study falls short of providing any heuristic for selecting among various possibilities for all communicable diseases. However, our hope is to highlight the key role of partial immunity assumptions, which remain seldom addressed.

## 4. Models and method

To address the question of the length of immunity in a real-world setting with a variable force of infection, we use non-Markovian discrete-time models. The overall idea behind non-Markovian models is to include residence times that depart from the exponential (and memoryless) waiting time assumption classically involved with constant rate processes in ODE-based models. Here we choose an approach that allows us to track individuals over time explicitly [5]. Using this approach, we may specify that individuals that recovered $a$ days age at time $t$, $R_{t,a}$ are subject to age-dependent processes, thereby allowing for reduced the force of infection in the leaky immunity assumption (*i.e.* a multiplying factor of $[1 - \xi(a)]$ on the force of infection) or an age-dependent rate to return to the susceptible compartment, $-\sigma(a)R_{t,a}$, in the polarised immunity. Additional details on the origin of non-Markovian models, the main different competing implementations that exist, and how to choose between them are available in Sect B in S1 Text.

## Modelling the initial waves of the Omicron variant

We implement two models focusing on SARS-CoV-2 hospital admissions data, one with leaky immunity and another one with polarised immunity. Our analysis focused on comparing the differences in the resulting epidemic dynamics on a long-term scale. Our focus is specifically on the Omicron period in France, beginning with its estimated introduction on November 26th, 2021 [49], and extending through to the end of official monitoring on January 31st, 2023. To simplify the initial situation, we assume a single age group and that everyone has already been immunised (either through infection-derived immunity or vaccination) by the time the Omicron BA.1 variant arrived. Although full immunisation was not achieved in practice, our aim is not to replicate the exact course of the French epidemic. Instead, we seek to highlight the significance of how immunity is defined within a plausible scenario. Depending on the immunity assumption, recovered individuals from Delta or previous strains or previously vaccinated ($R^{\mathrm{D}}$ compartment) may be infected directly by the Omicron strain (in the model with leaky immunity) or become susceptible ($S^{\mathrm{D}}$) before an Omicron infection (in the polarised approach). The $R^{\mathrm{D}}$ compartment is initialised with a uniform distribution on [0, 700] – individuals remaining until day 700 in $R^{\mathrm{D}}$ are considered entirely susceptible in both models. Being fully susceptible in both formalisms is either due to a decrease of susceptibility of 1 in the leaky model or going in the $S^d$ compartment in the polarised one (see Sect G in S1 Text).

Infected individuals are assumed to be divided into two groups, mild infections ($I$) and severe infections that always lead to hospitalisation ($J$), following a probability of hospitalisation ($p_h$) that we estimate from the data. Mildly infected individuals are assumed to recover directly and end up in the recovered compartment ($R^{\mathrm{o}}$). Severely infected individuals are hospitalised ($H$) before recovering while death was not modelled for simplicity. The delay distributions between mild infection and recovery, severe infection and hospital admission, and between hospital admission and recovery are parameterised using values from the literature ([4], as detailed in Sect E in S1 Text). Once in the recovered compartments, individuals can be reinfected directly in the leaky case, with a decrease in susceptibility that depends on the time spent in the recovered state. In the polarised model, recovered individuals become susceptible ($S^{\mathrm{o}}$) before getting reinfected. Both models are summarised in Fig 3.

In our model, Omicron evades acquired immunity (infection-derived or vaccination) with a given and fixed function based on empirical data from [11], as detailed in Sect E in S1 Text. Still building on the data from [11], we assume that the Omicron BA.1 strain induces an imperfect immunity that allows reinfection by the Omicron BA.1 strain (*cf.* Fig C in S1 Text). Finally, we do not consider the following Omicron sub-variants.

Our time series of interest is the daily hospital admissions, which is publicly available from the French public health agency (Santé Publique France). We use a Bayesian approach similar to that described by [50] to fit the piece-wise transmission rates, the initial proportion of Omicron-infected individuals, and the probability of hospitalisation that match the Omicron-induced daily new hospitalisations. The number of newly hospitalised individuals on day $t$ in the polarised model is given by

$$h_t^{\mathrm{polarised}} = p_h \sum_{k=0}^{t} \zeta_j \Lambda_{t-k} \left( S_{t-k}^{\mathrm{D}} + S_{t-k}^{\mathrm{o}} \right), \qquad (3)$$

where $p$ is the probability of being hospitalised, $\Lambda_t$ is (an approximation of) the force of infection on the day $t$, $\zeta(\cdot)$ the delay distribution between infection and hospitalisation and $S_t^{\mathrm{x}}$ the

individuals for which the strain $x$ infection-acquired immunity waned. In the leaky model, we have

$$h_t^{\text{leaky}} = p_h \sum_{k=0}^{t} \zeta_k \Lambda_{t-k} \sum_{i=0}^{N} \left( \left[ 1 - \xi_i^{\text{D}} \right] R_{t-k,i}^{\text{D}} + \left[ 1 - \xi_i^{\text{O}} \right] R_{t-k,i}^{\text{O}} \right), \tag{4}$$

where $\xi_i^x$ is the decrease in susceptibility that benefits individuals previously infected by strain $x$ when exposed to Omicron $i$ days after immunity acquisition, $R_{k,i}^x$ is the number of recovered individuals from strain $x$ on day $j$, $i$ days after their immunity acquisition. Let $N$ be the support of the recovered compartments. Note in both models, following the method developed in [5], we approximate the force of infection with

$$\Lambda_t = \frac{1}{1 + \left( \mathcal{R} c_t \sum_{i=0} \omega_i (I+J)_{t-i,i} \right)^{-1}}, \tag{5}$$

where $\mathcal{R}$ can be interpreted as the Omicron-era baseline reproduction number, $c_t$ is the transmission rate on day $t$, $\omega_i$ the generation time distribution and $(I+J)_{t,i}$ the number of infected individuals since $i$ days on day $t$. If the leaky and polarised immunity assumptions share the same formula, the force of infection is different in each model as the transmission rate $c_t$ is fitted for each model. Particularly, we use a Bayesian framework to estimate the transmission rate every 20 days (see Sect F in S1 Text). All model equations are available in Sect D in S1 Text and details on the fitting procedure are provided in Sect F in S1 Text.

## Sensitivity analysis

We perform a sensitivity analysis on the initialisation of the $R^{\text{D}}$ support to check the robustness of the result regardless of how we initialise the compartment. We test three other initialisations with different modes on the recovered support that ensure different levels of initial immunity given the immunity efficacy waning function.

We also perform a sensitivity analysis on the 20-day timespan window for the transmission rates to ensure it is not an artificial by-product to reproduce the observed dynamic. We also performed an out-of-sample validation on Google Mobility Data time series.

The results are available in Sect G in S1 Text.

## Supporting information

**S1 Text. Supplementary information.**
(PDF)

## Acknowledgments

We thank the ETE modelling team and Olivier Supplisson for useful discussions.

## Author contributions

**Conceptualization:** Bastien Reyné, Ramsès Djidjou-Demasse, Samuel Alizon, Mircea T. Sofonea.

**Data curation:** Bastien Reyné, Mircea T. Sofonea.

**Formal analysis:** Bastien Reyné, Mircea T. Sofonea.

**Investigation:** Bastien Reyné.

**Methodology:** Bastien Reyné, Tsukushi Kamiya, Ramsès Djidjou-Demasse, Mircea T. Sofonea.

**Supervision:** Ramsès Djidjou-Demasse, Samuel Alizon, Mircea T. Sofonea.

**Visualization:** Bastien Reyné.

**Writing – original draft:** Bastien Reyné.

**Writing – review & editing:** Bastien Reyné, Tsukushi Kamiya, Ramsès Djidjou-Demasse, Samuel Alizon, Mircea T. Sofonea.

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
