## [Decision Letter · Decision Letter 0]

23 Feb 2025

 PCOMPBIOL-D-25-00077

Leaky or polarised immunity: non-Markovian modelling highlights the impact of immune memory assumptions

PLOS Computational Biology

Dear Dr. Reyné,

Thank you for submitting your manuscript to PLOS Computational Biology. After careful consideration, we feel that it has merit but does not fully meet PLOS Computational Biology's publication criteria as it currently stands. Therefore, we invite you to submit a revised version of the manuscript that addresses the points raised during the review process.

All three reviewers appreciated the importance of the topic, however raised some concerns regarding technical choices and interpretation of the results. Further, the reviewers highlighted that various crucial aspects are somewhat hidden in the appendix, hence strongly suggest to restructure the manuscript in order to improve the reading flow and present all main aspects in the main body of the paper. In addition, various clarifications and language improvements are needed. Please find below the three detailed reviewer reports.

Please submit your revised manuscript within 60 days. If you will need more time than this to complete your revisions, please reply to this message or contact the journal office at ploscompbiol@plos.org. Please include the following items when submitting your revised manuscript:

We look forward to receiving your revised manuscript.

Kind regards,

Katharina Kusejko

Academic Editor

PLOS Computational Biology

Thomas Leitner

Section Editor

PLOS Computational Biology

**Journal Requirements:**

3) We notice that your supplementary Figures, and information are included in the manuscript file. Please remove them and upload them with the file type 'Supporting Information'. Please ensure that each Supporting Information file has a legend listed in the manuscript after the references list.

Potential Copyright Issues:

i) Figures 1A, and 1B. Please confirm whether you drew the images / clip-art within the figure panels by hand. If you did not draw the images, please provide (a) a link to the source of the images or icons and their license / terms of use; or (b) written permission from the copyright holder to publish the images or icons under our CC BY 4.0 license. Alternatively, you may replace the images with open source alternatives. See these open source resources you may use to replace images / clip-art:

**Reviewers' comments:**

Reviewer's Responses to Questions

**Comments to the Authors:**

**Please note that two reviews are uploaded as attachments.**

Reviewer #1: The manuscript investigates an important and interesting question in infectious disease modeling: How does the specific formalism used to describe immunity waning influence model dynamics/outcomes. The authors compare the leaky formalism (recovered individuals become more and more susceptible to reinfection as time since last infection/vaccination increases) and the polarized formalism (recovered individuals are fully immune but become fully susceptible over time). These two formalisms are frequently used in infectious disease models. For COVID-19 and many other diseases, the former formalism appears to be more useful but polarized immunity cannot be completely discounted. The authors display a great understanding of mathematical epidemiology and nicely motivate the choice of modeling framework. That being said, the most interesting discussions and answers related to the research question are in the various appendices (I am not sure why). The large number of appendices negatively affects the flow of the paper. Moreover, this manuscript suffers from poor English. While I realize that the authors are French and likely lack a native English speaker, a lot of the typos and grammar issues could have been prevented by the use of same English language software, e.g. Microsoft Word. Further, wording is at times confusing and the brevity of many explanations (i.e., lack of details) makes it hard to follow along.

Most importantly, I have severe concerns about the research findings. 1. The claim “the time between reinfections is always shorter with a leaky formalism” appears to be obvious given the way the authors implemented the two formalisms, specifically given the assumption of constant force of infection and in the absence of disease dynamics. See major comment 1 below. 2. The COVID-19 results appear to be largely confounded by assumptions made about initial conditions, which are not described anywhere, so I can only assume that the authors assumed everyone was recently recovered with 0 days since last immunity event at the onset of the Omicron waves, clearly a wrong assumption that affects the comparison between the leaky and polarized formalism. See major comment 2 below.

Major comments:

1. The authors should provide more details in the results section, clarifying the assumptions underlying the finding “the time between reinfections is always shorter with a leaky formalism”, specifically focusing on the validity of the strong assumption of a constant force of infection. Upon reading the Methods in detail I came to the conclusion that the results shown in Fig S1 in support of the claim “the time between reinfections is always shorter with a leaky formalism” are in a way a “self-fulfilling prophecy/circular conclusion”: The authors use the exact same function f in both formalisms. The polarized formalism assumes individuals that “lose immunity” become first susceptible, at which point they are susceptible to infection. The leaky formalism assumes individuals are always vulnerable to infection but a decreased rate. In other words, individuals are moved into the S and I compartment at the same rate, parametrized by the function f, in the polarized and leaky formalism, respectively. Naturally, this means reinfections occur earlier in the leaky formalism. We don’t need a model to obtain this result, as long as the force of infection is assumed to be constant and the same for the two formalisms. Note: the authors are aware of this, and derive this result, somewhat confusingly and with many assumptions, in Appendix E.

2. Figure 2 is very nice and helpful. It would be nice if the authors could add further details to the main text (Lines 91-94), specifically related to the initial values used in the models: at the beginning of the omicron wave (when the authors assume everyone has attained some immunity through infection or vaccination), how is the population distributed in terms of age of immunity in R^D_{t=0,a}, and what proportion, in the polarized formalism, starts in S^D? This strongly affects the outcomes! I could not find any information regarding this. Side note: Not everyone will have been infected with Delta. I would change “Last infection by Delta VOC” to something like “Immunity due to non-Omicron infection or vaccination”. See and modify also lines 220-225.

3. The 95% posterior trajectories intervals look identical for the leaky and the polarized formalism. Are the authors sure that this is correct and not a copy paste error? Given differences in the number of reinfections between the formalisms, I would expect to see differences in hospital admissions as well. Also, the total number of reinfections appears to be almost double in the leaky vs all-or-nothing formalism (3B) but the differences in daily infections (3C) are much smaller. This is strange, or am I missing something? Lastly, the authors should explain in the Results what exactly the “probability of hospitalization” refers to.

4. The appendix should not include new results. All results and supplementary figures should be referred to in the main manuscript, e.g., the global sensitivity analyses.

Minor comments:

L17: It would be nice if the authors could expand a little on “conservative assumptions”, either rewrite or add some more info.

L44/45: The sentence for the leaky vaccine is wrong. The decrease in susceptible is of size x, not 1-x. I.e., if x=80%, then the susceptibility of vaccinated/recovered individuals is 1-x = 20% compared to that of non-vaccinated/non-recovered people.

Fig 1C: R_N in black on dark blue cannot be read.

L86: awkward wording: “hardly remains generalisable” - is it generalizable or not? are the multiple reinfections the issue or the dynamical (i.e., non-constant force of infection) setting?

L155: a bimodal distribution of immune efficacy. It is unclear how this ties in with using antibody levels as correlates of protection. I hope the authors can clarify this in the text by rewriting/expanding this paragraph.

L504-514: I do not get the purpose of this paragraph. Using the strict definition of non-Markovian, yes, an SEIR model can be seen as non-Markovian. However, the way an SEIR model is typically implemented (using a fixed transition rate from E to I) it is indeed Markovian. The authors should clarify their point. Especially, because according to lines 528-530, the standard SEIR model would be Markovian even according to the authors.

616 “proportion of individualS that HAVE become …” - it’s important to

Appendix G: It would be nice to see plots/statistics that exhibit the goodness of fit between the hypo-exponential and Gamma distributions.

Sobol’s indices should be followed by a reference. Also, the information provided related to the sensitivity analysis is insufficient to reproduce the results.

Language:

L4: "and" not "or"

L14/15: "made the ...", grammar

L25: p missing: susceptible

L26/27: “It is the case of the first studies” - strange wording, not sure what exactly this means.

L45/46: if x in [0,1], get rid of % signs everywhere here. Or let x in [0,100] but that is the worse choice.

L47: delete to in “or to natural”

Fig 1 caption: simplest, not most simple

L64: assesS

L63: strange wording “This is partic- ularly the case of the study by”

L65: strange wording “as it should be studied further”

l74: “using simple modelS” or “using A simple model”

L128: “can fo unnoticed”???

L147: strange wording: “never discontinued to assess “

l170: strange use of “verifies”: we introduce the following function that verifies for all a

Fig 2 caption: the only difference … IS the following:

L212: witH

L215: not sure what this is supposed to mean: “We implement the two types of immunity described to SARS-CoV-2 incidence data“

L222: to illustrate, not illustrated

L433: grammar: “in every cases”

l433: I would add the word partial: “we assume partial immunity to last… “

L438: strange wording: “which is an order of magnitude observed quite frequently “

L480: replace “will” by “ability”?

L482: strange wording: “It will manifest itself through a wide variety of approaches over time”. E.g., why is future tense used?

l489: replace “may allow using of” by “enables the use of the”

L552: “implying a manual Euler scheme” is confusing. How about (if this is what the authors mean): “requiring the implementation of a manual Euler scheme”

l552: replace “way more easy” by “easier”

L558: awkward: “the choice goes to the modeller who will choose”

L566: become, no s

L570: “become infected becomeS”

577: “SAME” not smae

581: noticeD

585: would may ??

586: “The used Equation” - which equation? Why capitalize equation?

587: grammar: “as it do no involve”

607: re-infected, other places: reinfected - be consistent

609: I would add: “fully recovered (I.E., FULLY IMMUNE) before being …”

623: strange wording: “move also on the age of immunity support as time flies”

642: t_2

654: delete “we made”

655: Model equations

677: from of

This list is by no means exclusive. Any English language program, e.g. Microsoft Word, would have caught >50% of these errors.

Reviewer #2: Kindly refer to the attached report for details.

Reviewer #3: The review is uploaded as an attachment.

**Have the authors made all data and (if applicable) computational code underlying the findings in their manuscript fully available?**

Reviewer #1: Yes

Reviewer #2: None

Reviewer #3: Yes

PLOS authors have the option to publish the peer review history of their article (what does this mean?). If published, this will include your full peer review and any attached files.

Reviewer #1: No

Reviewer #2: No

Reviewer #3: No

**Figure resubmission:**
---

## [Decision Letter · Decision Letter 1]

4 Jun 2025

PCOMPBIOL-D-25-00077R1

Leaky or polarised immunity: non-Markovian modelling highlights the impact of immune memory assumptions

PLOS Computational Biology

Dear Dr. Reyné,

Thank you for submitting your manuscript to PLOS Computational Biology. After careful consideration, we feel that it has merit but does not fully meet PLOS Computational Biology's publication criteria as it currently stands. Therefore, we invite you to submit a revised version of the manuscript that addresses the points raised during the review process.

Please submit your revised manuscript within 30 days. If you will need more time than this to complete your revisions, please reply to this message or contact the journal office at ploscompbiol@plos.org. Please include the following items when submitting your revised manuscript:

We look forward to receiving your revised manuscript.

Kind regards,

Katharina Kusejko

Academic Editor

PLOS Computational Biology

Thomas Leitner

Section Editor

PLOS Computational Biology

**Journal Requirements:**

1) Please provide a completed 'Competing Interests' statement, including any COIs declared by your co-authors. If you have no competing interests to declare, please state "The authors have declared that no competing interests exist".

2) Please include the authors' affiliations in the online submission form. Please ensure that the affiliations of the authors listed on the manuscript title page do exactly match with the affiliations provided in the online submission form

NOTE: Affiliations should include a department (if applicable), an institution, a city, and a country.

**Reviewers' comments:**

Reviewer's Responses to Questions

**Comments to the Authors:**

**Please note that two reviews are uploaded as attachments.**

Reviewer #1: The authors took the comments and concerns from me and other reviewers seriously and the revised manuscript is substantially improved. I just have a few language suggestions.

Language:

l113: delete "epsilon(tau) where"

l162: replace "divergences" by "differences"

l192: "a unique" not "an unique"

Note: all my line number references are based on the marked up manuscript page numbers.

Reviewer #2: Please refer to the attached file.

Reviewer #3: The review is uploaded as an attachment.

**Have the authors made all data and (if applicable) computational code underlying the findings in their manuscript fully available?**

Reviewer #1: Yes

Reviewer #2: Yes

Reviewer #3: Yes

PLOS authors have the option to publish the peer review history of their article (what does this mean?). If published, this will include your full peer review and any attached files.

Reviewer #1: **Yes: **Claus Kadelka

Reviewer #2: No

Reviewer #3: **Yes: **Anna Bot

**Figure resubmission:**
---

## [Decision Letter · Decision Letter 2]

9 Jul 2025

PCOMPBIOL-D-25-00077R2

Leaky or polarised immunity: non-Markovian modelling highlights the impact of immune memory assumptions

PLOS Computational Biology

Dear Dr. Reyné,

Thank you for submitting your manuscript to PLOS Computational Biology. We received additional comments on the revised version of your manuscript, mostly concerning clarifying parameter choices and some inconsistencies with the notations, as well as some smaller adjustments. Therefore, we invite you to submit a revised version of the manuscript that addresses the points raised.

Please submit your revised manuscript within 30 days. If you will need more time than this to complete your revisions, please reply to this message or contact the journal office at ploscompbiol@plos.org. Please include the following items when submitting your revised manuscript:

We look forward to receiving your revised manuscript.

Kind regards,

Katharina Kusejko

Academic Editor

PLOS Computational Biology

Thomas Leitner

Section Editor

PLOS Computational Biology

**Reviewers' comments:**

Reviewer's Responses to Questions

**Comments to the Authors:**

**Please note that the review is uploaded as an attachment.**

Reviewer #2: Please refer to the attached document

**Have the authors made all data and (if applicable) computational code underlying the findings in their manuscript fully available?**

Reviewer #2: None

PLOS authors have the option to publish the peer review history of their article (what does this mean?). If published, this will include your full peer review and any attached files.

Reviewer #2: No

**Figure resubmission:**
---

## [Editor Report · Decision Letter 3]

6 Aug 2025

Dear Dr. Reyné,

We are pleased to inform you that your manuscript 'Leaky or polarised immunity: non-Markovian modelling highlights the impact of immune memory assumptions' has been provisionally accepted for publication in PLOS Computational Biology.

Best regards,

Katharina Kusejko

Academic Editor

PLOS Computational Biology

Thomas Leitner

Section Editor

PLOS Computational Biology

---

## [Editor Report · Acceptance letter]

PCOMPBIOL-D-25-00077R3

Leaky or polarised immunity: non-Markovian modelling highlights the impact of immune memory assumptions

Dear Dr Reyné,

I am pleased to inform you that your manuscript has been formally accepted for publication in PLOS Computational Biology. Your manuscript is now with our production department and you will be notified of the publication date in due course.

With kind regards,

Zsofia Freund
